# ProteinInvBench: Benchmarking Protein Inverse Folding on Diverse Tasks, Models, and Metrics

**Zhangyang Gao** [1,2,†]**, Cheng Tan** [1,2,†]**, Yijie Zhang** [3]**, Xingran Chen** [4]**, Lirong Wu** [1,2]**, Stan Z. Li** [2,*]

[1] Zhejiang University
[2] AI Lab, Research Center for Industries of the Future, Westlake University
[3] McGill University, [4] University of Michigan

## Abstract

Protein inverse folding has attracted increasing attention in recent years. However, we observe that current methods are usually limited to the CATH dataset and the recovery metric. The lack of a unified framework for ensembling and comparing different methods hinders the comprehensive investigation. In this paper, we propose ProteinInvBench, a new benchmark for protein design, which comprises extended protein design tasks, integrated models, and diverse evaluation metrics. We broaden the application of methods originally designed for single-chain protein design to new scenarios of multi-chain and *de novo* protein design. Recent impressive methods, including GraphTrans, StructGNN, GVP, GCA, AlphaDesign, ProteinMPNN, PiFold and KWDesign are integrated into our framework. In addition to the recovery, we also evaluate the confidence, diversity, sc-TM, efficiency, and robustness to thoroughly revisit current protein design approaches and inspire future work. As a result, we establish the first comprehensive benchmark for protein design, which is publicly available at `https://github.com/A4Bio/OpenCPD`.

## 1 Introduction

Protein inverse folding is a fundamental problem in biology and has many applications in medicine, agriculture, and bioenergy [1–4]. It has thereby attracted increasing attention in both the machine learning and biology communities [5, 6]. Traditional physical-inspired methods suffer from the problem of expensive computation and unsatisfactory accuracy. Recently, deep learning methods have shown great potential in science areas [7–30], including protein inverse folding, by simplifying the process and improving accuracy [12, 31–62]. Among them, we observe that graph-based methods achieve state-of-the-art performance. However, previous methods are usually limited to the CATH dataset and the recovery metric. We emphasize that recovery is not the only important metric for protein design. Other metrics such as confidence, diversity, TM-score, efficiency, and robustness are also important for comprehensively revisiting current approaches. Also, the evaluation dataset should be further extended from CATH dataset to broader or more difficult cases to facilitate practical applications. All these challenges motivate us to establish ProteinInvBench, a unified benchmark for protein inverse folding, in which multiple tasks, models, and metrics are introduced and integrated.

ProteinInvBench extends the task of protein design from single-chain to multi-chain and *de novo* protein design. To our knowledge, many computational protein design methods [41, 46, 47, 50, 63] have only been evaluated on the outdated single-chain structure dataset CATH4.2. Furthermore, few studies [48, 56] have investigated protein design performance in multi-chain tasks, and even fewer have evaluated the performance comparison on *de novo* protein structures. To fill this knowledge gap,

---

[†]Equal Contribution, [*]Corresponding Author.

37th Conference on Neural Information Processing Systems (NeurIPS 2023) Track on Datasets and Benchmarks.

we first benchmark open-sourced graph-based models on the latest CATH4.3 dataset and extend them to the case of multi-chain protein design. We then collect *de novo* protein structures with little or no similarity to existing structures from the CASP15 competition. Evaluating models on the CASP15 dataset allows us to gain a better understanding of the potential of AI models in designing *de novo* proteins and reveals that different models exhibit non-trivial differences in generalizability. We hope that the more complex and challenging tasks in ProteinInvBench will facilitate the development of protein design methods for practical applications.

ProteinInvBench also provides a range of metrics to comprehensively understand the strengths and weaknesses of each method. Beyond the recovery score that measures the percentage of exactly matched residues, we also evaluate the confidence, diversity, sc-TM, efficiency, and robustness. These metrics will be introduced in detail in the Sec.5. Notably, we use novel metrics such as confidence and sc-TM to measure the quality of designed sequences in an unsupervised and unbiased manner, respectively. With additional metrics, we encourage researchers to develop more robust, efficient models that can generate diverse proteins for higher success rates in wet experiments.

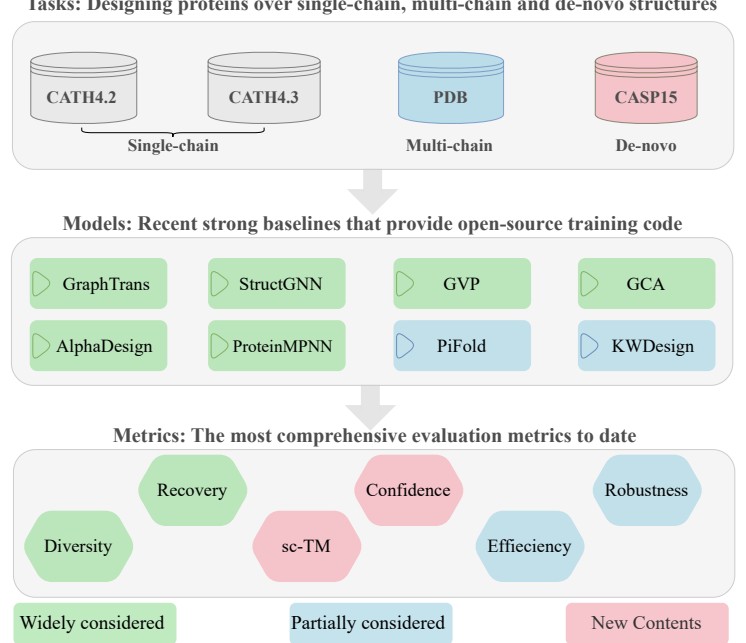

**Figure 1:** The framework of the proposed benchmark. The benchmark is organized incrementally from tasks to models, to metrics. We color contents in green and blue that are widely and partially considered by previous studies, respectively. Newly introduced contents are colored in pink.

Based on the constructed benchmark, we conduct extensive experiments to extend baseline models to new tasks and evaluate them on diverse metrics. All models are reproduced, integrated, and extended under a unified code framework. ProteinInvBench not only reproduces the reported results of baselines but also provides new insights into the detailed strengths and weaknesses of each method under different scenarios. To summarize, our contributions are as follows:

1. **Tasks:** We extend recent impressive models from single-chain protein design to the scenarios of multi-chain and *de novo* protein design.

2. **Models:** We integrate recent impressive models into a unified framework for efficiently reproducing and extending them to custom tasks.

3. **Metrics:** We incorporate new metrics such as confidence, sc-TM, and diversity for protein design, and integrate metrics including recovery, robustness, and efficiency to formulate a comprehensive evaluation system.

4. **Benchmark:** We establish the first comprehensive benchmark of protein design, providing insights into the strengths and weaknesses of different methods.

## 2    Overall Framework and Problem Definition

**Overall Framework**    We present the overall framework in Figure 1, which contains three components: (1) *Multiple datasets* for the task of single-chain, multi-chain, and de-novo protein design. (2) *Strong baselines* are integrated into our unified framework, including GraphTrans, GVP, GCA, AlphaDesign, ProteinMPNN, PiFold, and KWDesign. (3) *Diverse metrics* are used to evaluate the designed proteins in a quantitative and comprehensive manner.

**Problem Definition**    The protein inverse folding problem [64] aims to find the amino acids sequence $\mathcal{S} = \{s_i : 1 \leq i \leq n\}$ folding into the desired structure $\mathcal{X} = \{X_i \in \mathbb{R}^{m,3} : 1 \leq i \leq n\}$, where $m$ is the maximum number of points belonging to the $i$-th residue, $n$ is the number of residues and the natural proteins are composed by 20 types of amino acids, i.e., $1 \leq s_i \leq 20$ and $s_i \in \mathbb{N}^+$. Therefore, the protein inverse folding problem is usually formulated as a structure-to-sequence learning problem, where the goal is to learn a function $\mathcal{F}_\theta$:

$$\mathcal{F}_\theta : \mathcal{X} \mapsto \hat{\mathcal{S}}. \tag{1}$$

The function $\mathcal{F}_\theta$ is usually parameterized by a neural network, and the parameters $\theta$ are learned by minimizing the cross-entropy loss, i.e., $\mathcal{L}(\mathcal{F}_\theta(\mathcal{X}), \mathcal{S}) = -\sum_{i=1}^{n} \log s_i p(\hat{s}_i | \mathcal{X}, \theta)$

## 3    Datasets

**CATH**    The CATH (Class, Architecture, Topology, Homology) [65] database is a comprehensive resource for protein structure classification that hierarchical group proteins based on their structural features. The database defines classes based on topological similarities, architectures based on the arrangement of secondary structure elements, topologies based on the connectivity of secondary structure elements, and homologous domains based on sequence similarity. Previous works such as GraphTrans [50], GVP [41], and GCA [46] have used the CATH4.2 version of the database as a benchmark for protein design, which splits the dataset by CATH topology classification and includes 18,024 proteins for training, 608 proteins for validation, and 1,120 proteins for testing. However, CATH4.2 is an outdated version and may not represent the current protein structure space. To address this, we use the newer version, CATH4.3 for benchmarking protein design and follow the data splitting protocol of ESMIF [53]. This results in a training set of 16,153 structures, a validation set of 1,457 structures, and a test set of 1,797 structures. Note that the curated CATH dataset contains only single-chain structures and does not consider the case of designing multi-chain proteins.

**PDB**    The Protein Data Bank (PDB) [66] is a comprehensive database of 3D structural data for biological molecules. To study multi-chain protein design, we utilized a ProteinMPNN dataset derived from PDB assemblies with high resolution and less than 10,000 residues. The dataset was preprocessed by clustering sequences at 30% identity, resulting in 25,361 clusters. Following ProteinMPNN's setup, we divided the clusters randomly into training (23,358), validation (1,464), and test sets (1,539), ensuring that none of the chains from the target chain or biounits of the target chain were present in the other two sets. During each training epoch, we cycled through the sequence clusters and randomly selected a sequence member from each cluster. This dataset was used for the task of multi-chain protein design, expanding the comparison of computational protein design methods, as many previous methods were omitted in this task.

**TS45**    In addition to designing single- and multi-chain proteins, we also include a set of *de novo* proteins collected from the CASP15 competition to provide a more realistic assessment [67, 68]. The Critical Assessment of Protein Structure Prediction (CASP15), which took place from May through August 2022, was held after the release dates of CATH4.3 (July 1, 2019) and PDB (August 2, 2021). In CASP15, diverse protein targets are introduced, including FM (Free Modeling), TBM (Template-Based Modeling), TBM-easy, and TBM-hard proteins. There are 18 FM, 25+2 TBM (including 20 TBM-eazy, 5 TMB-hard, 2 FM/TBM). The FM targets have no homology to any known protein structure, making them particularly suitable for *de novo* protein design. The TBM targets have some homology to known protein structures, while the TBM-easy targets are relatively easy TBM targets. The TBM-hard targets are more difficult TBM targets, with lower levels of sequence identity to known structures. We download the public TS-domains structures from CASP15 which consists of 45 structures, namely TS45. We use TS45 as a benchmark for *de novo* protein design, as the structures are less similar to known structures and were not determined prior to the construction of the training sets.

# 4 Baseline models

We ensemble recent strong protein design baselines under the unified framework. Currently, we support open-sourced graph methods, such as GraphTrans, StructGNN [50], GVP [41], GCA [46], AlphaDesign [47], ProteinMPNN [48], PiFold [63] and KWDesign [69], that we can access their codes and training scripts. StructGNN and GraphTrans [50] employ C-alpha for geometric features, while GCA [46] adds global attention. GVP [41] introduces a novel GNN layer for invariant and equivariant features. However, these methods suffer from poor inference efficiency due to autoregressive decoders. To overcome this, AlphaDesign [47] replaces the decoder with an iterative 1D CNN. Recent advancements include ProteinMPNN [48], which incorporates additional structural information, and PiFold [63], a combination of AlphaDesign and ProteinMPNN. KWDesign [69] is an ensemble model that utilizes PiFold to create a prompt template. It refines the template using pre-trained knowledge, including sequence pretraining (ESM-650M [70]) and structure pretraining models (ESMIF's encoder [53]).

New baselines [56, 71] will continue to be added in the future. We have not included ESMIF [53] in our benchmark since it lacks a training script and is challenging for us to train. We plan to add it to the benchmark once we successfully train the model. According to the generation scheme, these baselines can be categorized as autoregressive, iterative, and one-shot models.

**Autoregressive models** consider both sequential and structural dependencies by combining the structural encoder and autoregressive sequence decoder, such as GraphTrans [50], GVP [41], GCA [46], and ProteinMPNN [48]. The protein sequences are generated by:

$$p(\hat{\mathcal{S}}|\mathcal{X};\theta) = \prod_{t=1}^{n} p(\hat{s}_t|\hat{s}_{<t}, \mathcal{X};\theta). \tag{2}$$

Autoregressive models have been criticized for being slow in generating long proteins [63].

**Iterative models** generate residues in parallel and iteratively refine the generated sequence (AlphaDesign [47] and KWDesign [69]):

$$\hat{\mathcal{S}}^{(0)} \sim p(\hat{\mathcal{S}}|\mathcal{X};\theta^{(0)}), \hat{\mathcal{S}}^{(t)} \sim p(\hat{\mathcal{S}}|\hat{\mathcal{S}}^{(t-1)}, \mathcal{X};\theta^{(t)}), \tag{3}$$

where $t$ indicates the refinement step, affecting the inference time costs. $\theta$ is a learnable parameter.

**One-shot models** generate the protein sequence in parallel, e.g., PiFold [63], which is quite efficient in generating long proteins, written as:

$$\hat{\mathcal{S}} \sim p(\hat{\mathcal{S}}|\mathcal{X};\theta). \tag{4}$$

# 5 Metrics

In this section, we introduce metrics that will be used for protein design evaluation, including recovery, confidence, diversity, and sc-TM. Previous researches [41, 46, 47, 50, 53, 63] mainly focus on improving recovery, while ignoring other metrics. However, we argue that recovery is not the only important metric for protein design. Other metrics introduced follows are also crucial for comprehensively revisiting current approaches, such as confidence, diversity, sc-TM, efficiency, and robustness.

**Recovery** Recovery is the primary metric of the ability of the designed protein to recover its original residues, defined as:

$$\texttt{Rec} = \frac{1}{n} \sum_{i=1}^{n} \mathbb{1}(\hat{s}_i = s_i) \tag{5}$$

where $\mathbb{1}(\cdot)$ is the indicator function, $\hat{s}_i$ is the designed residue at the $i$-th position, and $s_i$ is the corresponding reference residue. A high recovery rate indicates that the designed protein sequence is similar to the reference sequence, and it is therefore expected that the folded structure will also be similar to the reference structure. Since measuring structural similarity is computationally expensive, previous protein design methods have placed a great deal of emphasis on improving recovery. However, it is important to note that recovery itself is a proxy metric for measuring structural similarity. In other words, a higher recovery rate does not necessarily ensure a higher level of structural similarity. Moreover, a high recovery rate may result in low diversity.

**Confidence** Calculating recovery requires access to the reference sequence, which is not always available in practice. When the ground-truth sequence is unknown, measuring and ranking the quality of the designed sequence becomes more challenging. We introduce the confidence metric to address this problem, which is the average predictive probability of designed amino acids, defined as:

$$\texttt{Conf} = \frac{1}{n} \sum_{i=1}^{n} p(\hat{s}_i) \tag{6}$$

**Diversity** To improve the success rate of protein design, it is important to explore a set of protein sequences rather than placing a bet on a single sequence. In this case, generating diverse sequences is crucial for exploring the reasonable protein sequence space. We define the pairwise diversity [72] as $D_{ij} = \frac{\sum_{l=1}^{n} \mathbb{1}_{r_{i,l} \neq r_{j,l}}}{n}$, where $r_{i,l}$ indicates the $l$-th residue of the $i$-th designed sequence. The overall diversity score is

$$\texttt{Div} = \sum_{i,j} \frac{D_{i,j}}{m^2} \tag{7}$$

where $i, j \in \{1, 2, 3, \cdots, m\}$ and $m$ is the number of totally designed sequences. By default, we set $m = 10$. However, measuring diversity alone without combining it with other metrics may be misleading. For example, a high diversity indicates a low recovery rate, more likely to result in a low structural similarity.

**sc-TM** The structural similarity is the ultimate standard for measuring the quality of the designed sequence. However, the structures of designed protein sequences needed to be predicted by other algorithms, such as AlphaFold [73], RoseTTAFold [74], OmegaFold [75] and ESMFold [70]. The protein folding algorithm itself has a certain inductive bias and will cause some prediction errors, which will affect the evaluation. To overcome the inductive bias, we introduce the self-consistent TM-score (sc-TM) metric:

$$\texttt{sc-TM} = \texttt{TMScore}(f(\hat{\mathcal{S}}), f(\mathcal{S})) \tag{8}$$

where $f$ is the protein folding algorithm and $\texttt{TMscore}(\cdot, \cdot)$ is a widely used metric [76] for measuring protein structure similarity. Since the structures of the designed sequence and reference sequence are predicted by the same protein folding algorithm, the model's inductive bias is expected to be canceled out when calculating the TM-score. This approach results in a more robust metric, called the sc-TM, that is less affected by the inductive bias of the protein folding algorithm.

**Robustness** Robustness measures an algorithm's ability to maintain its original performance under geometric perturbations. It is a useful metric for assessing the stability and generalizability of an algorithm. We define robustness as:

$$\texttt{Rob} = \texttt{Rec}' - \texttt{Rec} \tag{9}$$

where $\texttt{Rec}$ and $\texttt{Rec}'$ are the recovery after and before applying small Gaussian perturbations to the Cartesian coordinates of the structure, correspondingly. As the template protein structures may not be perfect, more robust methods are expected to be more suitable in real-world applications.

**Efficiency** Efficiency measures the computational resources and time required to design a set of proteins. This study reports the training time, evaluation time, and model parameters of different methods over the standard benchmarks. While efficiency may not be a crucial problem compared to the recovery and sc-TM, it is a useful metric for assessing the model's scalability and practicality.

## 6 Benchmarking Protein Design

In this section, we retrain baselines on the newly introduced datasets and evaluate them using diverse metrics, resulting in a comprehensive benchmark. The experiments are organized as follows:

1. **Establish a basic benchmark within recovery and confidence.** As emphasized by previous studies, the recovery rate and predictive confidence are the most important and straightforward metrics. We benchmark baselines over these metrics on CATH4.2, CATH4.3, PDB, and TS45 for the task of single-chain, multi-chain, and *de novo* protein design, respectively. These results could serve as the basic benchmark for future studies.

2. **Measuring diversity and sc-TM for practical challenging tasks in protein design.** We further extend the evaluation metrics to diversity and sc-TM. The diversity is opposite of recovery and is meaningless if we measure it alone. By examining the sequence diversity and structural sc-TM together, we could have a more comprehensive understanding of the designable protein space.

3. **Assessing the robustness when input structures are not perfect.** Although the model performs well on natural proteins, it may fail when the artificially designed structure is noisy. In this case, the robustness of the model is crucial for practical applications. We evaluate the robustness of different methods by applying geometric perturbations to the template protein structures during the evaluation phase.

4. **Comparing the efficiency.** Towards designing efficient, scalable, and generalizable algorithms, we evaluate the efficiency in terms of training time, evaluation time, and model parameters to facilitate the development of more efficient protein design methods.

### 6.1 Recovery and Confidence

In this section, we benchmark the recovery rate and confidence of different methods on the CATH4.2, CATH4.3, PDB, and TS45 datasets to address the problems of single-chain, multi-chain, and *de novo* protein design. By extending from CATH4.2 to the newer CATH4.3 and from single-chain to multi-chain to *de novo*, we have constructed the most comprehensive benchmark to date for protein design. All models are retrained and evaluated under the same code framework for a fair comparison. The hyperparameters used for training models are provided in the Appendix.

**Table 1:** Single-chain results. The **best** and suboptimal results are labeled with bold and underlined.

| | Model length | Confidence ↑ | | | | Recovery % ↑ | | | |
|---|---|---|---|---|---|---|---|---|---|
| | | $L < 100$ | $100 \leq L < 300$ | $300 \leq L < 500$ | Full | $L < 100$ | $100 \leq L < 300$ | $300 \leq L < 500$ | Full |
| CATH4.2 | StructGNN | 0.31 | 0.45 | 0.45 | 0.43 | 0.26 | 0.36 | 0.36 | 0.35 |
| | GraphTrans | 0.31 | 0.43 | 0.43 | 0.43 | 0.25 | 0.35 | 0.35 | 0.34 |
| | GCA | 0.34 | 0.46 | 0.47 | 0.45 | 0.27 | 0.38 | 0.38 | 0.37 |
| | GVP | 0.40 | 0.52 | 0.53 | 0.51 | 0.28 | 0.40 | 0.41 | 0.39 |
| | AlphaDesign | 0.36 | 0.49 | 0.49 | 0.47 | 0.33 | 0.43 | 0.44 | 0.42 |
| | ProteinMPNN | 0.38 | 0.51 | 0.52 | 0.50 | 0.32 | 0.47 | 0.47 | 0.45 |
| | PiFold | 0.44 | 0.58 | 0.60 | 0.57 | 0.39 | 0.53 | 0.56 | 0.52 |
| | **KWDesign** | **0.50** | **0.68** | **0.72** | **0.67** | **0.44** | **0.62** | **0.66** | **0.61** |
| CATH4.3 | StructGNN | 0.35 | 0.41 | 0.47 | 0.41 | 0.30 | 0.34 | 0.40 | 0.34 |
| | GraphTrans | 0.37 | 0.42 | 0.48 | 0.42 | 0.29 | 0.34 | 0.39 | 0.34 |
| | GCA | 0.38 | 0.43 | 0.49 | 0.43 | 0.32 | 0.36 | 0.41 | 0.36 |
| | GVP | 0.45 | 0.51 | 0.55 | 0.50 | 0.33 | 0.38 | 0.45 | 0.38 |
| | AlphaDesign | 0.41 | 0.48 | 0.53 | 0.47 | 0.37 | 0.43 | 0.47 | 0.42 |
| | ProteinMPNN | 0.42 | 0.49 | 0.57 | 0.49 | 0.38 | 0.44 | 0.52 | 0.44 |
| | PiFold | 0.47 | 0.56 | 0.64 | 0.55 | 0.43 | 0.52 | 0.59 | 0.51 |
| | **KWDesign** | **0.58** | **0.68** | **0.76** | **0.67** | **0.51** | **0.61** | **0.69** | **0.60** |

**Single-chain Results** The results of single-chain protein design are shown in Tab. 1, where both CATH4.2 and CATH4.3 datasets are included. We present metrics for proteins of different sequence lengths. From Tab. 1, it could be concluded that:

1. KWDesign and PiFold are the best and second-best models. They consistently outperform all other models in terms of both confidence and recovery across all protein lengths in both CATH4.2 and CATH4.3 datasets. This highlights their effectiveness towards protein inverse folding.

2. Models perform better on longer proteins. This could be due to the increased complexity and information available for longer proteins, allowing the models to make more confident predictions.

3. CATH4.2 and CATH4.3 datasets show the same performance trend and very similar results, thereby validating the performance consistency of the different models. However, it also informs us that they are unable to provide complementary information for analyzing protein design methods. More diverse, complex, and challenging datasets are needed for further investigation.

4. The unsupervised confidence is highly correlated to supervised recovery. This discovery suggests that researchers can rank the quality of designed proteins based on confidence alone, without needing to access the ground truth.

**Multi-chain Results** To remedy the problem that CATH4.2 and CATH4.3 are highly consistent and do not bring complementary information, we extend the experiment to a multi-chain dataset. The corresponding results are presented in Table 2, showing that:

**Table 2:** Multi-chain results. The **best** and suboptimal results are labeled with bold and underlined.

| Model length | | Confidence ↑ | | | | Recovery % ↑ | | | |
|---|---|---|---|---|---|---|---|---|---|
| | | $L < 100$ | $100 \leq L < 500$ | $500 \leq L < 1000$ | Full | $L < 100$ | $100 \leq L < 500$ | $500 \leq L < 1000$ | Full |
| PDB | StructGNN | 0.49 | 0.49 | 0.50 | 0.49 | 0.41 | 0.41 | 0.42 | 0.41 |
| | GraphTrans | 0.48 | 0.47 | 0.48 | 0.48 | 0.40 | 0.39 | 0.40 | 0.40 |
| | GCA | 0.45 | 0.45 | 0.46 | 0.45 | 0.41 | 0.41 | 0.42 | 0.41 |
| | GVP | 0.51 | 0.53 | 0.55 | 0.54 | 0.44 | 0.42 | 0.45 | 0.43 |
| | AlphaDesign | 0.52 | 0.53 | 0.54 | 0.53 | 0.48 | 0.49 | 0.50 | 0.49 |
| | ProteinMPNN | 0.54 | 0.56 | 0.58 | 0.57 | 0.52 | 0.53 | 0.55 | 0.53 |
| | PiFold | 0.56 | 0.60 | 0.63 | 0.61 | 0.54 | 0.58 | 0.60 | 0.58 |
| | **KWDesign** | **0.65** | **0.71** | **0.74** | **0.71** | **0.59** | **0.66** | **0.67** | **0.66** |

1. KWDesign achieves the best performance across all the models for proteins of all lengths, followed by PiFold and ProteinMPNN.

2. The longer the protein sequence, the higher the recovery. Like the single-chain case, confidence and recovery generally increase with the length of the protein chain. As the length of multi-chain protein could be up to 1000, models perform better on the PDB than on the CATH dataset.

*De novo* **Protein Design**  To investigate the models' potential in designing novel proteins, we evaluate pre-trained models on TS45. We present the *de novo* protein design results in Tab. 3, considering four subsets of TS45: FM (Free Modeling), TBM (Template-Based Modeling), TBM-easy, and TBM-hard. The results show that:

1. For models pre-trained on CATH4.2 and CATH4.3, KWDesign outperforms others by a large margin. The PiFold model consistently performs as the second-best model after KWDesign.

2. For models pre-trained on PDB, PiFold achieves the best performance, while ProteinMPNN provides very competitive recoveries. Switching from the CATH to the PDB dataset, ProteinMPNN achieves a more significant performance gain than PiFold.

3. The consistent performance trend across different protein subsets suggests that the difficulty level of the protein design task depends on the nature of the protein subset. For instance, models tend to perform better on TBM-easy proteins than on TBM-hard proteins. A more challenging subset of proteins may help reveal the shortcomings of current protein design algorithms.

4. AI methods have demonstrated great potential in *de novo* protein design, with all models (except StructGNN and GraphTrans) achieving recoveries of approximately 40% or higher. However, there is still slight performance degradation compared to the results on the original test set.

**Table 3:** Results of *de novo* protein design. The **best** and suboptimal results are labeled with bold and underlined.

| Training set | Model | Confidence ↑ | | | | | Recovery % ↑ | | | | |
|---|---|---|---|---|---|---|---|---|---|---|---|
| | | FM | TBM | TBM-eazy | TBM-hard | Full | FM | TBM | TBM-eazy | TBM-hard | Full |
| CATH4.2 | StructGNN | 0.41 | 0.43 | 0.48 | 0.43 | 0.45 | 0.35 | 0.33 | 0.38 | 0.35 | 0.35 |
| | GraphTrans | 0.39 | 0.43 | 0.46 | 0.42 | 0.44 | 0.33 | 0.30 | 0.37 | 0.36 | 0.36 |
| | GCA | 0.48 | 0.43 | 0.53 | 0.48 | 0.50 | 0.39 | 0.37 | 0.41 | 0.38 | 0.40 |
| | GVP | 0.48 | 0.49 | 0.50 | 0.50 | 0.49 | 0.37 | 0.33 | 0.42 | 0.39 | 0.39 |
| | AlphaDesign | 0.44 | 0.41 | 0.50 | 0.46 | 0.48 | 0.41 | 0.36 | 0.46 | 0.41 | 0.42 |
| | ProteinMPNN | 0.49 | 0.48 | 0.53 | 0.51 | 0.52 | 0.44 | **0.41** | 0.46 | 0.40 | 0.44 |
| | PiFold | 0.52 | 0.46 | 0.59 | 0.53 | 0.55 | 0.47 | 0.38 | 0.50 | 0.47 | 0.47 |
| | KWDesign | **0.55** | **0.52** | **0.70** | **0.62** | **0.64** | **0.49** | 0.40 | **0.59** | **0.55** | **0.54** |
| CATH4.3 | StructGNN | 0.40 | 0.40 | 0.45 | 0.43 | 0.44 | 0.35 | 0.33 | 0.38 | 0.37 | 0.36 |
| | GraphTrans | 0.39 | 0.42 | 0.46 | 0.43 | 0.45 | 0.35 | 0.32 | 0.37 | 0.35 | 0.35 |
| | GCA | 0.46 | 0.42 | 0.51 | 0.44 | 0.48 | 0.37 | 0.33 | 0.43 | 0.40 | 0.41 |
| | GVP | 0.47 | 0.45 | 0.50 | 0.48 | 0.49 | 0.37 | 0.31 | 0.41 | 0.38 | 0.39 |
| | AlphaDesign | 0.44 | 0.40 | 0.50 | 0.47 | 0.48 | 0.40 | 0.36 | 0.44 | 0.44 | 0.42 |
| | ProteinMPNN | 0.49 | 0.48 | 0.53 | 0.49 | 0.52 | 0.44 | 0.34 | 0.48 | 0.43 | 0.46 |
| | PiFold | 0.54 | 0.45 | 0.56 | 0.51 | 0.54 | 0.47 | 0.38 | 0.52 | 0.49 | 0.49 |
| | KWDesign | **0.59** | **0.50** | **0.70** | **0.63** | **0.65** | **0.50** | **0.43** | **0.59** | **0.60** | **0.56** |
| PDB | StructGNN | 0.46 | 0.41 | 0.53 | 0.47 | 0.48 | 0.39 | 0.34 | 0.42 | 0.41 | 0.41 |
| | GraphTrans | 0.43 | 0.42 | 0.51 | 0.45 | 0.48 | 0.38 | 0.33 | 0.44 | 0.40 | 0.41 |
| | GCA | 0.45 | 0.41 | 0.49 | 0.45 | 0.47 | 0.40 | 0.33 | 0.44 | 0.43 | 0.43 |
| | GVP | 0.51 | 0.46 | 0.55 | 0.53 | 0.53 | 0.40 | 0.32 | 0.47 | 0.43 | 0.43 |
| | AlphaDesign | 0.49 | 0.43 | 0.54 | 0.50 | 0.51 | 0.43 | 0.39 | 0.48 | 0.46 | 0.46 |
| | ProteinMPNN | 0.56 | 0.49 | 0.58 | 0.55 | 0.55 | 0.52 | 0.39 | 0.55 | 0.51 | 0.52 |
| | PiFold | 0.55 | 0.48 | 0.59 | 0.53 | 0.57 | 0.52 | 0.45 | 0.53 | 0.52 | 0.53 |
| | KWDesign | **0.60** | **0.67** | **0.69** | **0.65** | **0.66** | **0.56** | **0.59** | **0.60** | **0.62** | **0.59** |

## 6.2 Diversity and sc-TM

**Diversity** We benchmark the diversity on TS45 dataset using models pre-trained on CATH4.3. As discovered by previous research [48, 53], the sampling temperature affects diversity. Denote the temperature as $T$, the predicted probability vector is $\boldsymbol{p} \in \mathbb{R}^{n,20}$, we sample new sequences from the distribution of $\texttt{Multinomial}(\texttt{softmax}(\boldsymbol{p}/T))$. We vary the temperature from 0.0 to 0.5 and plot the trends of recovery and diversity in Fig. 2. Under the same sampling temperature, high recovery leads to decreased diversity. However, at the same level of recovery, stronger models have higher diversity.

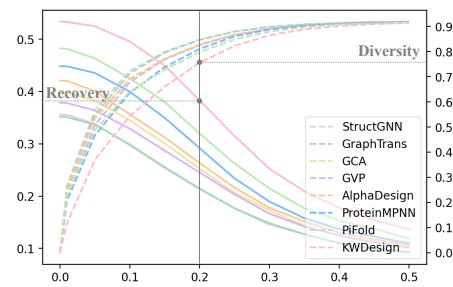

**Figure 2:** The trends of recovery and diversity.

**sc-TM** While generating diverse protein sequences is appealing, it would be meaningless if these sequences could not fold to structures with topologies similar to the reference one. With this in mind, we investigate the recovery and sc-TM metrics as the temperature increases. To compute sc-TM, we utilize AlphaFold2 [73] to predict protein structures from sequences. According to Fig.3 and Fig.2, we observe that a slight increase in temperature from 0 to 0.1 is beneficial in significantly enhancing diversity while maintaining good recovery and sc-TM. However, increasing the temperature to 0.5 renders the designed sequences meaningless in recovery and sc-TM, despite the higher diversity. Carefully tuning the temperature would be beneficial in practical applications.

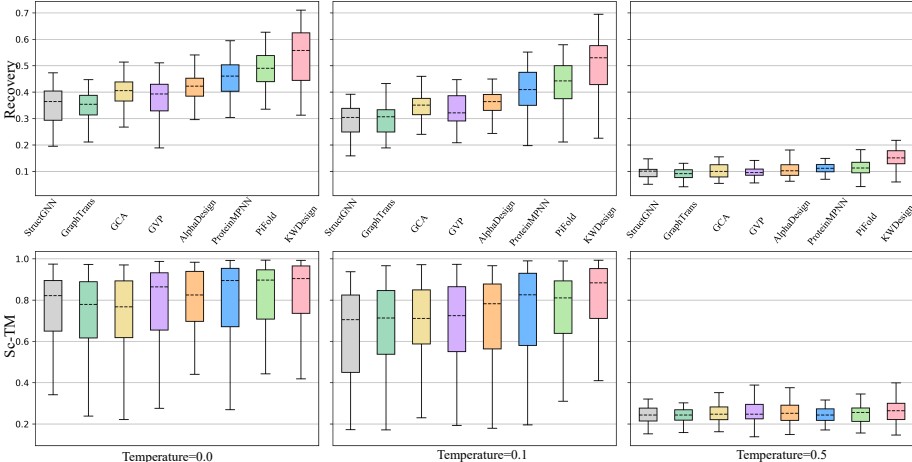

**Figure 3:** The statistics of recovery and sc-TM with increasing temperature.

## 6.3 Robustness and Efficiency

**Robustness** We further investigate whether the models are robust to structure perturbations, as the artificially designed structures may not be perfect, and the atom position may deviate slightly due to thermodynamic vibrations or errors in experimental measurements. We add different Gaussian noise to the input structure, i.e., $\mathcal{X} \leftarrow \mathcal{X} + \epsilon \mathcal{N}(0, I)$. Note that the Gaussian noise (in Angstrom) is added in both training and evaluation structures, where the noise scale $\epsilon$ is chosen from $[0.02, 0.2, 0.5, 1.0]$. As we have shown that models perform consistently on CATH and PDB, we benchmark the robustness on CATH4.3. The experimental results are shown in Tab.4, from which we observe that:

1. Weaker models tend to exhibit greater robustness than stronger models. For example, with $\epsilon = 1$, the recoveries of StructGNN and GraphTrans decrease by only 14%, while AlphaDesign, ProteinMPNN, PiFold, and KWDesign decrease by at least 20%. This is a natural outcome, as weaker models may be the first to reach the performance floors of the dataset.

2. KWDesign achieves the highest recovery across noise scales, followed by PiFold. StructGNN, GraphTrans, GCA, and GVP degrade quickly and reach similar lower bounds. AlphaDesign is more affected by noise compared to GCA and GVP, likely due to its reliance on angular features, which are more sensitive to noise than distance features.

3. All models show a decrease in performance as the Gaussian noise scale increases. Developing protein design methods with higher robustness remains challenging.

**Table 4:** Results of robustness. We calculate the difference in terms of model performance on the noisy and clean structures and show it in parentheses. A smaller absolute value of this difference indicates a more robust model. The **best** and suboptimal robust models are labeled with bold and underlined.

| Model length | Confidence ↑ | | | | Recovery % ↑ | | | |
|---|---|---|---|---|---|---|---|---|
| | $L < 100$ | $100 \leq L < 300$ | $300 \leq L < 500$ | Full | $L < 100$ | $100 \leq L < 300$ | $300 \leq L < 500$ | Full |
| **CATH4.3** ($\epsilon = 0.02$) StructGNN | 0.37 (+0.02) | **0.42 (+0.01)** | 0.49 (+0.02) | 0.42 (+0.01) | 0.28 (-0.02) | **0.33 (-0.01)** | 0.38 (-0.02) | **0.33 (-0.01)** |
| GraphTrans | **0.37 (+0.00)** | 0.41 (-0.01) | **0.48 (+0.00)** | 0.41 (-0.01) | **0.28 (-0.01)** | 0.32 (-0.02) | 0.37 (-0.02) | 0.32 (-0.02) |
| GCA | 0.36 (-0.02) | 0.40 (-0.03) | 0.47 (-0.02) | 0.41 (-0.02) | 0.29 (-0.03) | 0.33 (-0.03) | 0.39 (-0.02) | 0.33 (-0.03) |
| GVP | 0.44 (-0.01) | 0.48 (-0.03) | 0.54 (-0.01) | 0.51 (+0.01) | 0.29 (-0.04) | 0.34 (-0.04) | 0.43 (-0.02) | 0.36 (-0.02) |
| AlphaDesign | 0.42 (+0.01) | 0.50 (+0.02) | 0.56 (+0.03) | 0.49 (+0.02) | 0.33 (-0.04) | 0.39 (-0.04) | 0.43 (-0.04) | 0.38 (-0.04) |
| ProteinMPNN | 0.41 (-0.01) | 0.47 (-0.02) | 0.55 (-0.02) | 0.46 (-0.03) | 0.32 (-0.06) | 0.40 (-0.04) | 0.49 (-0.03) | 0.40 (-0.04) |
| PiFold | 0.41 (-0.06) | 0.51 (-0.05) | 0.60 (-0.04) | 0.49 (-0.06) | 0.37 (-0.06) | 0.47 (-0.05) | **0.54 (-0.02)** | 0.45 (-0.06) |
| KWDesign | 0.50 (-0.08) | 0.63 (-0.05) | 0.72 (-0.04) | 0.61 (-0.06) | 0.43 (-0.08) | 0.56 (-0.05) | 0.65 (-0.04) | 0.54 (-0.06) |
| **CATH4.3** ($\epsilon = 0.2$) StructGNN | **0.34 (-0.01)** | **0.36 (-0.05)** | **0.41 (-0.06)** | **0.36 (-0.05)** | 0.25 (-0.05) | **0.28 (-0.06)** | **0.32 (-0.08)** | **0.28 (-0.06)** |
| GraphTrans | 0.33 (-0.04) | 0.36 (-0.06) | 0.39 (-0.09) | 0.36 (-0.06) | **0.25 (-0.04)** | 0.27 (-0.07) | 0.31 (-0.08) | 0.27 (-0.07) |
| GCA | 0.33 (-0.05) | 0.35 (-0.08) | 0.39 (-0.10) | 0.35 (-0.08) | 0.25 (-0.07) | 0.28 (-0.08) | 0.31 (-0.10) | 0.28 (-0.08) |
| GVP | 0.39 (-0.06) | 0.43 (-0.08) | 0.45 (-0.10) | 0.42 (-0.08) | 0.25 (-0.08) | 0.28 (-0.10) | 0.34 (-0.11) | 0.29 (-0.09) |
| AlphaDesign | 0.35 (-0.06) | 0.40 (-0.08) | 0.43 (-0.10) | 0.39 (-0.08) | 0.29 (-0.08) | 0.33 (-0.10) | 0.36 (-0.11) | 0.33 (-0.09) |
| ProteinMPNN | 0.37 (-0.05) | 0.41 (-0.08) | 0.47 (-0.10) | 0.41 (-0.08) | 0.29 (-0.09) | 0.35 (-0.09) | 0.41 (-0.11) | 0.35 (-0.09) |
| PiFold | 0.35 (-0.12) | 0.43 (-0.13) | 0.48 (-0.16) | 0.41 (-0.14) | 0.32 (-0.09) | 0.39 (-0.13) | 0.45 (-0.14) | 0.39 (-0.12) |
| KWDesign | 0.43 (-0.15) | 0.53 (-0.15) | 0.60 (-0.16) | 0.52 (-0.15) | 0.38 (-0.13) | 0.47 (-0.14) | 0.54 (-0.15) | 0.46 (-0.14) |
| **CATH4.3** ($\epsilon = 0.5$) StructGNN | **0.30 (-0.05)** | **0.31 (-0.10)** | **0.34 (-0.13)** | **0.31 (-0.10)** | 0.22 (-0.08) | **0.24 (-0.10)** | 0.26 (-0.14) | **0.24 (-0.10)** |
| GraphTrans | 0.30 (-0.07) | 0.31 (-0.11) | 0.33 (-0.15) | 0.31 (-0.11) | **0.22 (-0.07)** | 0.23 (-0.11) | 0.25 (-0.14) | **0.24 (-0.10)** |
| GCA | 0.30 (-0.08) | 0.31 (-0.12) | 0.34 (-0.15) | 0.31 (-0.12) | 0.22 (-0.10) | 0.24 (-0.12) | 0.26 (-0.15) | 0.24 (-0.12) |
| GVP | 0.32 (-0.13) | 0.34 (-0.17) | 0.37 (-0.18) | 0.35 (-0.15) | 0.22 (-0.11) | 0.25 (-0.13) | 0.26 (-0.19) | 0.25 (-0.13) |
| AlphaDesign | 0.30 (-0.11) | 0.33 (-0.15) | 0.35 (-0.18) | 0.33 (-0.14) | 0.26 (-0.11) | 0.28 (-0.15) | 0.30 (-0.17) | 0.28 (-0.14) |
| ProteinMPNN | 0.34 (-0.08) | 0.36 (-0.13) | 0.39 (-0.18) | 0.37 (-0.12) | 0.26 (-0.12) | 0.29 (-0.15) | 0.33 (-0.19) | 0.30 (-0.14) |
| PiFold | 0.32 (-0.15) | 0.36 (-0.20) | 0.40 (-0.24) | 0.35 (-0.20) | 0.30 (-0.13) | 0.34 (-0.18) | 0.37 (-0.22) | 0.33 (-0.18) |
| KWDesign | 0.38 (-0.20) | 0.47 (-0.21) | 0.52 (-0.24) | 0.45 (-0.22) | 0.33 (-0.18) | 0.42 (-0.19) | **0.47 (-0.14)** | 0.41 (-0.19) |
| **CATH4.3** ($\epsilon = 1.0$) StructGNN | **0.27 (-0.08)** | **0.26 (-0.14)** | **0.28 (-0.19)** | **0.27 (-0.14)** | 0.19 (-0.11) | 0.20 (-0.14) | 0.21 (-0.19) | 0.20 (-0.14) |
| GraphTrans | 0.26 (-0.11) | 0.26 (-0.16) | 0.27 (-0.21) | 0.26 (-0.16) | **0.19 (-0.10)** | 0.19 (-0.15) | 0.20 (-0.19) | **0.20 (-0.14)** |
| GCA | 0.25 (-0.13) | 0.25 (-0.18) | 0.26 (-0.23) | 0.25 (-0.18) | 0.19 (-0.13) | 0.19 (-0.17) | 0.20 (-0.21) | 0.19 (-0.17) |
| GVP | 0.29 (-0.16) | 0.29 (-0.22) | 0.30 (-0.25) | 0.27 (-0.23) | 0.20 (-0.13) | 0.20 (-0.18) | 0.21 (-0.24) | 0.20 (-0.18) |
| AlphaDesign | 0.16 (-0.25) | 0.16 (-0.32) | 0.15 (-0.38) | 0.16 (-0.31) | 0.18 (-0.19) | 0.18 (-0.25) | 0.18 (-0.29) | 0.18 (-0.24) |
| ProteinMPNN | 0.31 (-0.11) | 0.30 (-0.19) | 0.32 (-0.25) | 0.31 (-0.18) | 0.22 (-0.16) | 0.23 (-0.21) | 0.25 (-0.27) | 0.23 (-0.21) |
| PiFold | 0.28 (-0.19) | 0.29 (-0.27) | 0.32 (-0.32) | 0.29 (-0.26) | 0.26 (-0.17) | 0.28 (-0.24) | 0.29 (-0.30) | 0.28 (-0.23) |
| KWDesign | 0.33 (-0.25) | 0.42 (-0.26) | 0.45 (-0.31) | 0.40 (-0.27) | 0.29 (-0.22) | 0.37 (-0.24) | 0.41 (-0.28) | 0.35 (-0.25) |

**Efficiency**    To encourage efficient and scalable models, we also benchmark the training cost, evaluation cost, the number of trainable parameters, and training epochs in Tab.5. We conclude that:

1. Training these models over a single epoch is generally fast, except for KWDesign (w/o memory). Fortunately, with the memory retrieval technique, KWDesign can achieve a similar speed as the other models. It is worth noting that PiFold and KWDesign require only up to 20 epochs to achieve competitive performance.

2. In terms of evaluation efficiency, iterative and one-shot models like AlphaDesign and PiFold perform exceptionally well, as they do not require autoregressive generation. On the other hand, KWDesign is relatively slower in this category as it needs to make multiple calls to large pre-trained models to generate higher-quality sequences.

3. Stronger models are associated with a higher number of trainable parameters. Among these models, GVP shows superior efficiency in utilizing model parameters. KWDesign achieves the best performance with the most parameters.

**Table 5:** Efficiency comparison.

| Model | Training Cost | | | Evaluation Cost | | | Others | |
|---|---|---|---|---|---|---|---|---|
| | CATH4.2 | CATH4.3 | PDB | CATH4.2 | CATH4.3 | PDB | Trainable Params | # epochs |
| StructGNN | **120s** | **112s** | 600s | 378s | 662s | 1068s | 1.38MB | 100 |
| GraphTrans | 130s | 123s | 583s | 438s | 737s | 1232s | 1.53MB | 100 |
| GCA | 660s | 604s | 1308s | 378s | 688s | 1020s | 2.09MB | 100 |
| GVP | 402s | 380s | 840s | 1874s | 3193s | 3753s | **0.93MB** | 100 |
| AlphaDesign | 290s | 267s | 546s | **31s** | **50s** | **75s** | 6.62MB | 100 |
| ProteinMPNN | 165s | 154s | **540s** | 347s | 570s | 889s | 1.66MB | 100 |
| PiFold | 410s | 364s | 780s | 39s | 69s | 162s | 5.79MB | 20 |
| KWDesign(w/o memory) | 3820s | 3624s | – | 451s | 752s | – | 54.49MB | 20 |
| KWDesign(w memory) | 453s | 437s | – | – | – | – | | 20 |

# 7    Conclusion

Protein inverse folding has received significant attention in recent years. However, the lack of thorough comparisons across multiple tasks and metrics hinders the progress toward practical applications. To address this issue, we propose ProteinInvBench, which consists of diverse tasks, models, and metrics and provides a comprehensive view of computational protein design. We plan to update ProteinInvBench when the CATH dataset (every 12 months) is updated.

## Acknowledgement

This work was supported by the National Key R&D Program of China (2022ZD0115100), the National Natural Science Foundation of China (U21A20427), the Competitive Research Fund (WU2022A009) from the Westlake Center for Synthetic Biology and Integrated Bioengineering.

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
