# A Appendix

**Hyper-parameters & Training Details** We use the default model hyperparameters, such as the number of layers and hidden dimensions, according to the original paper for each baseline. The tunable hyperparameters are learning rate, batch size and dropout rate. We use grid search to find the best hyperparameters for each model. The learning rate is chosen from $\{0.001, 0.0005, 0.0001, 0.00005, 0.00001\}$, the batch size is chosen from $\{8, 16, 32\}$, and the dropout rate is chosen from $\{0.0, 0.1, 0.2, 0.3, 0.4, 0.5\}$. We find that a learning rate of 0.001, a batch size of 8, and a dropout rate of 0.1 produce good results for most models. We train all models using the OneCycle scheduler and Adam optimizer for up to 100 epochs, except for PiFold and KWDesign. We observe that PiFold and KWDesign achieve satisfactory results when trained for only 20 epochs. The number of trainable parameters and training time for each model are shown in Table 5. The training time is measured on a single NVIDIA Tesla A100 GPU.

|  | GraphTrans | StructGNN | GVP | GCA | AlphaDesign | ProteinMPNN | PiFold | KWDesign |
|---|---|---|---|---|---|---|---|---|
| #GNN Layers | 6 | 6 | 6 | 6 | 10 | 6 | 10 | 10*3 |
| #Hidden dim | 128 | 128 | 128 | 128 | 128 | 128 | 128 | 128 |
| batch size | 8 | 8 | 2000 (max node) | 8 | 8 | 8 | 8 | 8 or 32 |

**Table 6:** Hyperparameters for each model. Note that KWDesign recycles three times, where ten layers of GNNs are used for each time. GVP has its unique batch sampling strategy, where the maximum number of nodes in a batch is 2000.

**Limitation & Boarder Impact** This research proposes a computational benchmark for structure-based protein design. However, we do not conduct wet experiments to further validate the results. The designed proteins may be unstable or not foldable in real-world applications. As with any new technology, there are safety concerns associated with the use of designed proteins. For example, using designed proteins as therapeutics may have unintended side effects, or releasing them into the environment may have unforeseen ecological impacts. Protein design raises ethical issues, particularly in designing proteins with potentially harmful or controversial functions. Nevertheless, we believe that protein design has great potential to benefit society and that negative impacts can be eliminated through industry regulation and legislation. The proposed benchmark is useful for the community to develop new protein design algorithms and to evaluate the performance of existing models.