# OpenReview forum: "ProteinInvBench: Benchmarking Protein Inverse Folding on Diverse Tasks, Models, and Metrics"
_NeurIPS.cc/2023/Track/Datasets_and_Benchmarks — NeurIPS 2023 Datasets and Benchmarks Poster_

### Official Review · Reviewer_Gd38 · 2023-06-30

**Rating:** 6
**Confidence:** 4
**Clarity:** See statement on Clarity under Streng…

**Strengths:**

## Significance and relevance to the broader research community
Predicting amino acid identities from structure is an important problem, arising both in protein design and in recent advances in generative modelling of proteins. A large number of new methods have come out in recent years, but it has been difficult to track exactly how significant the progress has been. In my opinion, this is therefore a useful resource - and I expect it will have an impact on the Bio-ML subcommunity at NeurIPS, and more broadly on the many downstream users of these methods.

## Quality of the research
The authors choose a number of datasets which are composed according to slightly different criteria. This makes it possible to contrast the performance of the different SOTA methods in interesting ways. Generally, I think these design choices are sound, and the results reveal meaningful differences between the methods.

## Clarity of paper
The paper is clearly structured, and easy to follow.

**Additional Feedback:**

Title: "ProteinBench"
Suggestion: There seem to be quite a few protein data benchmarks coming out at the moment (I’ve reviewed 3 just for this conference). The authors could therefore perhaps consider if it could be helpful to the community if they chose a name that reflects the type of data and tasks (for instance by including “Design” in the name)

Line 14. "available at https://github.com/A4Bio/OpenCPD."
There seems to be some inconsistency in naming of the benchmark. The documentation on GitHub (and the github repo itself) refers to it as opencpd, while the paper frees to it as ProteinBench.

Line 16-21.
The authors introduce the concept of "Structure based protein design" in general, and then discuss "deep learning methods have shown great potential", but right after that the scope seems to change from general structure-based protein design to the more specific task of predicting amino acid identities given a fixed structural environment. It would benefit the paper if this distinction was more clearly made - and if it was mentioned early on that this latter task is sometimes referred to as "inverse folding". When reading the manuscript, I was originally confused to think that the benchmark would cover a broader set of tasks than it did.

Line 71, "where n is the number of residues"
I found this notation a bit confusing. It seems to suggest that you only have a single atom coordinate per residue. Do you limit yourself to Calpha-only representations of protein structures? If so, this should be stated explicitly - since it is potentially a limitations if future SOTA methods require a more complete structural representation.

Line 96. "The dataset was preprocessed by clustering sequences at a 30% sequence identity"
How was this done for multiple chains? Did you test sequence identity as a whole or ensure max 30% per chain?

Line 113 "We download the public TS-domains structures from CASP15 which consists of 45 structures, namely TS45"
It was unclear how these 45 structures were distributed into the FM, TBM, TBM-easy, TBM-hard categories that you just introduced. Could you clarify?

Line 116 "were not determined prior to the construction of the training sets."
I agree that this must be true for CATH4.2. But are you sure that CATH4.3 and PDB do not include these structures (or homologues to them)?

Line 152. "Calculating recovery requires access to the reference sequence, which is not always available in practice."
I was confused by this statement. In which cases among your datasets is the reference sequence not available? It seems to me that in CATH and PDB the sequences are available, and that they are also provided for CASP targets. It also seems to me that you would need the sequence to measure the "Recovery" metric that you introduced in the previous paragraph. Or are you simply referring to potential real-life scenarios where you might wish to assess quality without having ground truth values available. Please clarify.

Line 152. Confidence.
In earlier work, the perplexity has been used as a metric as an alternative to the confidence score that you propose. It would be helpful if the authors could motivate in the paper why they opted for the Confidence metric instead - since the perplexity seems to me like a slightly richer metric (reflecting the peakedness of the entire distribution and not just the peakedness of the predicted label)

Line 177. "before applying perturbations"
Consider briefly stating here what type of perturbations these are. E.g. "before applying small Gaussian perturbations to the Cartesian coordinates of the structure". I realize that you specify this later on, but it would make it slightly easier to read if it was mentioned here.

Line 230. "Unfortunately, KWDesign crashed during the training..."
Since this is a benchmark, it is unfortunate that it includes such an inconclusive result. It would be useful if the authors could verify whether this is a fundamental flaw of KWDesign or a simple technical issue (which could potentially be solved by moving to a GPU will more memory).

Line 234. "As the length of multi-chain protein could be up to 1000, models perform better on the PDB than on the CATH dataset."
This could also be caused by the fact that you are splitting on sequence, rather than on structure, so you could imagine the PDB splits are “easier”

Figure 2.
This plot is very difficult to read. Could you use colors for the different methods instead?

Line 258 "good recovery and sc-TM."
You state “good sc-TM”. But given that the TM ranges between 0 and 1 the variance seems very high. Would it be possible to check if this was an artefact of the ESM fold method, and whether one of the other structure prediction methods would provide more robust results (e.g. just comparing it on a small subset with full AlphaFold predictions)

Figure 3.
The x-labels are placed in a strange position, and difficult to connect to the bars in the bottom plots. Perhaps use colors and a legend? Also, I would suggest using a shared y-axis between the plots, so that they can be directly compared.

Line 265. "Note that the Gaussian noise is added in both training and evaluation structures, where the noise scale ϵ is chosen from [0.02, 0.2, 0.5, 1.0]."
Please specify which unit this is in. Angstrom?


# Minor details
Line 20-21.
A lot of citations here, and it's cumbersome to look them up, because they are scattered all over the reference list. Perhaps use a citation style where citations are ordered according to order of occurrence - which would also have the benefit that you could just write [XX-YY]?

Line 69. "The structure based protein design" -> "Structure based protein design" or "The structure based protein design problem"

**Correctness:**

The experiments appear to be well-designed (see further details in Strengths above). The conclusions are well supported by the conducted experiments.

**Documentation:**

Code is available through the provided github link. However, the data itself seems to not be available (the link "To download the processed datasets, please click here" refers to an unrelated arxiv paper and "The processed datasets could be found in the releases" link is non-functional). There is also no hosting, licencing and maintance plan, which makes it difficult to assess the long-term perspectives of this dataset.

**Ethics:**

I have no ethical concerns about this paper.

**Limitations:**

The manuscript provides an appendix detailing potential limitations of the provided benchmark.

**Opportunities For Improvement:**

Although I have a list of questions and suggestions for improvement to the authors (see Additional Comments below), I generally have a positive impression of the paper. My main concern is the data availability from the provided website (dead links etc, see below) - in particular since this manuscript is submitted in the benchmark/dataset track. I am therefore giving it a below threshold score now, but am happy to increase the score if these issues are fixed during rebuttal.

**Relation To Prior Work:**

Although individual methods for inverse folding are discussed (and included in the benchmark), the authors do not explicitly discuss prior work on benchmarks for this (or related) tasks. For instance, the Atom3D dataset contains a Residue Identity task, which I think is identical to the task presented here, and should probably be mentioned (I am not affiliated with that work).

**Summary And Contributions:**

The manuscript introduces a new benchmark for structure-based protein design (i.e. inverse folding), allowing for systematic comparisons of the state of the art in the field. The benchmark includes the original CATH4.2 dataset, an updated CATH4.3 dataset, a PDB dataset and a de novo dataset consisting of structures from CASP without close homologues. The paper also introduces a broader set of metrics, moving beyond the the simple reconstruction/recovery metrix typically used.

---

> ### Author Response · Authors · 2023-08-17
> **Author rebuttal, part1**
>
> Dear reviewer,
>
> Thanks for your valuable and careful suggestions! Your comments will be helpful to help us improve the manuscript.
>
> > **W1** However, the data itself seems to not be available (the link "To download the processed datasets, please click here" refers to an unrelated arxiv paper and "The processed datasets could be found in the releases" link is non-functional).
> >> **Reply to W1** We released CATH4.2, CATH4.3, CASP15 datasets on out github. You can check out the right-hand plane of the project's GitHub page. Alternatively, you can download these data from https://drive.google.com/drive/folders/1O8pUC25XVdtuweAmvIaanOk8cSuW_a_q. The ProteinMPNN's PDB dataset could be downloaded from https://files.ipd.uw.edu/pub/training_sets/pdb_2021aug02.tar.gz
>
> > **Q1** Title: "ProteinBench" Suggestion: There seem to be quite a few protein data benchmarks coming out at the moment (I’ve reviewed 3 just for this conference). The authors could therefore perhaps consider if it could be helpful to the community if they chose a name that reflects the type of data and tasks (for instance by including “Design” in the name)
> >> **Reply to Q1** Adding "design" to "ProteinBench" seems to result in a long abbreviation. We have mentioned in the full title (Benchmarking Protein Design on Diverse Tasks, Models, and Metrics) to tell readers that we focus on the problem of protein design. We have changed "ProteinBench" to "ProteinInvBench", indicating the protein inverse folding benchmark.
>
> > **Q2** Line 14. "available at https://github.com/A4Bio/OpenCPD." There seems to be some inconsistency in naming of the benchmark. The documentation on GitHub (and the github repo itself) refers to it as opencpd, while the paper frees to it as ProteinBench.
> >> **Reply to Q2** We forgot to update the repository name when submitting the paper, which means "Open Computational Protein Design (OpenCPD)". We have aligned these names.
>
> > **Q3** Line 16-21. The authors introduce the concept of "Structure based protein design" in general, and then discuss "deep learning methods have shown great potential", but right after that the scope seems to change from general structure-based protein design to the more specific task of predicting amino acid identities given a fixed structural environment. It would benefit the paper if this distinction was more clearly made - and if it was mentioned early on that this latter task is sometimes referred to as "inverse folding". When reading the manuscript, I was originally confused to think that the benchmark would cover a broader set of tasks than it did.
> >> **Reply to Q3** Thanks for your good suggestion. We have replaced "Structure based protein design" with "Protein inverse folding".
>
> > **Q4** Line 71, "where n is the number of residues" I found this notation a bit confusing. It seems to suggest that you only have a single atom coordinate per residue. Do you limit yourself to Calpha-only representations of protein structures? If so, this should be stated explicitly - since it is potentially a limitations if future SOTA methods require a more complete structural representation.
> >> **Reply to Q4** Thanks for the good suggestions! We have revised the manuscript accordingly.
>
> > **Q5** Line 96. "The dataset was preprocessed by clustering sequences at a 30% sequence identity" How was this done for multiple chains? Did you test sequence identity as a whole or ensure max 30% per chain?
> >> **Reply to Q5** Actually, we adopt the curated dataset and splits following ProteinMPNN. We do not find whether they measure the sequence identity as a whole or ensure max 30% per chain. Therefore, we cannot answer this question.
>
> > **Q6** Line 113 "We download the public TS-domains structures from CASP15 which consists of 45 structures, namely TS45" It was unclear how these 45 structures were distributed into the FM, TBM, TBM-easy, TBM-hard categories that you just introduced. Could you clarify?
> >> **Reply to Q6** The FM, TBM, TBM-easy, TBM-hard categories are defined by CASP15. In TS45, there are 18 FM, 25+2 TBM (including 20 TBM-eazy, 5 TMB-hard, 2 FM/TBM).
>
> > **Q7** Line 116 "were not determined prior to the construction of the training sets." I agree that this must be true for CATH4.2. But are you sure that CATH4.3 and PDB do not include these structures (or homologues to them)?
> >> **Reply to Q7** CASP15 is holded after the release date of CATH4.3 and PDB. From May through August 2022, CASP organizers have been posting on this website sequences of unknown protein structures for modeling. CATH4.3 uses PDB proteins as of July 1, 2019. ProteinMPNN's PDB dataset is as of 2 August 2021.

---

> ### Author Response · Authors · 2023-08-17
> **Author rebuttal, part2**
>
> > **Q8** Line 152. "Calculating recovery requires access to the reference sequence, which is not always available in practice." I was confused by this statement. In which cases among your datasets is the reference sequence not available? It seems to me that in CATH and PDB the sequences are available, and that they are also provided for CASP targets. It also seems to me that you would need the sequence to measure the "Recovery" metric that you introduced in the previous paragraph. Or are you simply referring to potential real-life scenarios where you might wish to assess quality without having ground truth values available. Please clarify.
> >> **Reply to Q8** You are correct. We mean the potential real-life scenarios where we wish to assess quality without having ground truth values available.
>
> > **Q9** Line 152. Confidence. In earlier work, the perplexity has been used as a metric as an alternative to the confidence score that you propose. It would be helpful if the authors could motivate in the paper why they opted for the Confidence metric instead - since the perplexity seems to me like a slightly richer metric (reflecting the peakedness of the entire distribution and not just the peakedness of the predicted label)
> >> **Reply to Q9** Confidence could also be used for measuring the quality of each residue, while perplexity measures the quality of the whole sequence. As to real-life scenarios, experts can fix some residues with high confidence and mutate the residues with low confidence based on their domain knowledge.
>
> > **Q10** Line 177. "before applying perturbations" Consider briefly stating here what type of perturbations these are. E.g. "before applying small Gaussian perturbations to the Cartesian coordinates of the structure". I realize that you specify this later on, but it would make it slightly easier to read if it was mentioned here.
> >> **Reply to Q10** Thanks for your suggestion. We have revised the manuscript accordingly.
>
> > **Q11** Line 230. "Unfortunately, KWDesign crashed during the training..." Since this is a benchmark, it is unfortunate that it includes such an inconclusive result. It would be useful if the authors could verify whether this is a fundamental flaw of KWDesign or a simple technical issue (which could potentially be solved by moving to a GPU will more memory).
> >> **Reply to Q11** After persistent efforts, we identified that the collapse was attributed to the feature construction of quaternions. We have rewritten the feature construction code and retrained KWDesign accordingly. The manuscript has been updated to include the additional results.
>
> > **Q12** Line 234. "As the length of multi-chain protein could be up to 1000, models perform better on the PDB than on the CATH dataset." This could also be caused by the fact that you are splitting on sequence, rather than on structure, so you could imagine the PDB splits are “easier”
> >> **Reply to Q12** Thanks for your insightfull comments! We agree with you that this is a potential explanation.
>
> > **Q13** Figure 2. This plot is very difficult to read. Could you use colors for the different methods instead?
> >> **Reply to Q13** Thanks for your feedback! We have updated the figure with more colors. Under the same sampling temperature, high recovery leads to decreased diversity. However, at the same level of recovery, stronger models have higher diversity.  This phenomenon can be attributed to the fact that stronger models, such as KWDesign, PiFold, and ProteinMPNN, exhibit relatively high confidence levels for a larger number of residues. Even after applying smoothing to the probability distribution, the recovery rate remains consistently high, while a noticeable improvement is observed in terms of diversity.
>
> > **Q14** Line 258 "good recovery and sc-TM." You state “good sc-TM”. But given that the TM ranges between 0 and 1 the variance seems very high. Would it be possible to check if this was an artefact of the ESM fold method, and whether one of the other structure prediction methods would provide more robust results (e.g. just comparing it on a small subset with full AlphaFold predictions)
> >> **Reply to Q14** A good suggestion! The process of MSA searching incurs significant computational and time costs when utilizing AlphaFold and OpenFold. Currently, we are in the process of employing AlphaFold2 to evaluate the scTM. Once this analysis is complete, we will update the paper with the results obtained from AlphaFold2.
>
> > **Q15** Figure 3. The x-labels are placed in a strange position, and difficult to connect to the bars in the bottom plots. Perhaps use colors and a legend? Also, I would suggest using a shared y-axis between the plots, so that they can be directly compared.
> >> **Reply yo Q15** Thanks for your suggestions. We redraw the plots in different colors for each method with a shared y-axis. To minimize redundancy, we kept the x-axis shared by both plots.

---

> ### Author Response · Authors · 2023-08-17
> **Author rebuttal, part3**
>
> > **Q16** Line 265. "Note that the Gaussian noise is added in both training and evaluation structures, where the noise scale ϵ is chosen from [0.02, 0.2, 0.5, 1.0]." Please specify which unit this is in. Angstrom?
> >> **Reply to Q16** The unit is in Angstrom.
>
> > **Q17** Line 20-21. A lot of citations here, and it's cumbersome to look them up, because they are scattered all over the reference list. Perhaps use a citation style where citations are ordered according to order of occurrence - which would also have the benefit that you could just write [XX-YY]?
> >> **Reply to Q17** Thanks for your kind instructions, we will have a try.
>
> > **Q18** Line 69. "The structure based protein design" -> "Structure based protein design" or "The structure based protein design problem"
> >> **Reply to Q18** Thansks for your comments! We have revised the phase as "protein inverse folding problem".
>
> We appreciate all the efforts you have made to assist us in improving our manuscript. If our responses have satisfactorily addressed your concerns, we kindly request that you consider raising your score to support the acceptance of our work.
>
> Best,
>
> Authors.

---

> > ### Comment · Reviewer_Gd38 · 2023-08-23
> > **Response to rebuttal**
> >
> > Thanks to the authors for their rebuttal. Since the data availability issues are now resolved, I will increase my score to 6.
> >
> > There are still some minor issues remaining, which I encourage the authors to consider prior to camera-ready:
> >
> > * I noticed that one of the other reviewers requested a more thorough review of related work, and that the authors provided this in the rebuttal, but wrote that *"due to limited space, we cannot expand on the details"*. I just want to make sure that the authors are aware that they actually have en entire extra page available to them. From the "Message to NeurIPS 2023 Track Datasets and Benchmarks Authors" email:
> > > You can make revisions to your paper and supplementary materials, and you are allowed one additional page to address the reviewers’ comments. Ensure that it is easy for reviewers to find how their comments were addressed.
> >
> >     In my opinion, the discussion of the relevance of the different models would benefit the paper, and it would therefore be great if the elaborated discussion of related work provided in the rebutal could be included in the paper.
> >
> > * I appreciate that the authors have now made the datasets available, and solved the issue with the dead links. It seems that for the ProteinMPN dataset, the authors refer to data on the server of the original publication. Since this is a dataset paper, would it perhaps make sense for the authors to host all the data themselves, so they are not dependent on the original authors removing the data?
> >
> > * With respect to updates of the dataset, the authors state in the rebuttal to one of the other reviewers how they intend to do this regularly, but as far as I can see, this information is not present in the article itself (or the supporting material). It would be useful if a Maintenance Plan section was added to include this information.
> >
> > * There still seem to be some inconsistencies in naming: the title of the paper is now "ProteinInvBench", the title of the github repo is "ProteinBench", and the title on the README.md is "OpenCPD: Open-source Toolbox for Computaional Protein Design". Both OpenCPD and ProteinBench are used throughout the README file. Finally, the citation section in the README file seems to refer to a completely different paper.
> >
> > * Regarding Q6-Q7, I thank the reviewers for clarifying this to me in the rebuttal, but would encourage the authors to also add this information to the paper.
> >
> > * Regarding Q9. As far as I can see, you can define a perplexity per position (rather than the whole chain). There might be very good reasons for your choice, but since perplexity is used quite frequently in the literature it would be valuable if these reasons were provided in the paper - i.e. an explicit discussion of why confidence was chosen and perplexity was omitted.
> >
> > * Regarding Q16. Please add the information about the unit (Angstrom) to the manuscript in addition to providing it in this rebuttal. It's quite common to work with molecular distances in nanometers instead of Angstrom, so this is a relevant clarification.
> >
> > * Regarding Q18. You can for instance use the "unsrt" bibliography style to get this effect.

---

> > > ### Author Response · Authors · 2023-08-28
> > > **Further clarification, part1**
> > >
> > > Dear reviewer,
> > >
> > > Thanks for your careful review, constructive suggestions, and increased score! We are sincerely grateful for your review service. We provide further clarifications in terms of your additional comments.
> > >
> > > > **Q1+** I noticed that one of the other reviewers requested a more thorough review of related work, and that the authors provided this in the rebuttal, but wrote that "due to limited space, we cannot expand on the details". I just want to make sure that the authors are aware that they actually have en entire extra page available to them. From the "Message to NeurIPS 2023 Track Datasets and Benchmarks Authors" email: "You can make revisions to your paper and supplementary materials, and you are allowed one additional page to address the reviewers’ comments. Ensure that it is easy for reviewers to find how their comments were addressed." In my opinion, the discussion of the relevance of the different models would benefit the paper, and it would therefore be great if the elaborated discussion of related work provided in the rebutal could be included in the paper.
> > > >> **Reply to Q1+** Thanks for your suggestion! We are not sure if the additional pages will be counted in the main body of the manuscript or if they are simply for rebuttal, since the submission is limited to 9 pages of content. Nevertheless, we discuss the relevance of the different models in lines 118-126.
> > >
> > > > **Q2+** I appreciate that the authors have now made the datasets available, and solved the issue with the dead links. It seems that for the ProteinMPN dataset, the authors refer to data on the server of the original publication. Since this is a dataset paper, would it perhaps make sense for the authors to host all the data themselves, so they are not dependent on the original authors removing the data?
> > > >> **Reply to Q2+** Thanks for your careful comments! We will upload ProteinMPNN dataset to https://www.idrive.com/idrive/sh/sh?k=p9b2y3l6i5, which would take some time.
> > >
> > > > **Q3+** With respect to updates of the dataset, the authors state in the rebuttal to one of the other reviewers how they intend to do this regularly, but as far as I can see, this information is not present in the article itself (or the supporting material). It would be useful if a Maintenance Plan section was added to include this information.
> > > >> **Reply to Q3+** We have added the Maintenance Plan in the conclusion section.
> > >
> > > > **Q4+** There still seem to be some inconsistencies in naming: the title of the paper is now "ProteinInvBench", the title of the github repo is "ProteinBench", and the title on the README.md is "OpenCPD: Open-source Toolbox for Computaional Protein Design". Both OpenCPD and ProteinBench are used throughout the README file. Finally, the citation section in the README file seems to refer to a completely different paper.
> > > >> **Reply to Q4+** Thank you for your feedback! We have made the necessary changes to the GitHub repository and updated the title on the README.md file to "ProteinInvBench". As we have not yet uploaded the paper to arXiv, the updated citation will be provided at a later time.
> > >
> > > > **Q5+** Regarding Q6-Q7, I thank the reviewers for clarifying this to me in the rebuttal, but would encourage the authors to also add this information to the paper.
> > > >> **Reply to Q5+** We have added this information in lines 103-107.

---

> > > ### Author Response · Authors · 2023-08-28
> > > **Further clarification, part2**
> > >
> > > > **Q6+** Regarding Q9. As far as I can see, you can define a perplexity per position (rather than the whole chain). There might be very good reasons for your choice, but since perplexity is used quite frequently in the literature it would be valuable if these reasons were provided in the paper - i.e. an explicit discussion of why confidence was chosen and perplexity was omitted.
> > > >> **Reply to Q6+** Thanks for your comments! Given the limited range of the confidence level, which is typically between 0 and 1, it is more convenient to set a threshold value. We have evaluated all methods based on the confidence score, which could serve as a widely-used measure.
> > >
> > > > **Q7+** Regarding Q16. Please add the information about the unit (Angstrom) to the manuscript in addition to providing it in this rebuttal. It's quite common to work with molecular distances in nanometers instead of Angstrom, so this is a relevant clarification.
> > > >> **Reply to Q7+** We have updated the information in line 272.
> > >
> > > > **Q8+** Regarding Q18. You can for instance use the "unsrt" bibliography style to get this effect.
> > > >> **Reply to Q8+** Thanks for your help! The "\bibliographystyle{unsrtnat}" works well.
> > >
> > > > **Q14**  Line 258 "good recovery and sc-TM." You state “good sc-TM”. But given that the TM ranges between 0 and 1 the variance seems very high. Would it be possible to check if this was an artefact of the ESM fold method, and whether one of the other structure prediction methods would provide more robust results (e.g. just comparing it on a small subset with full AlphaFold predictions)
> > > >> **Additional reply to Q14** As promised, we have updated Figure 3 with the results obtained from AlphaFold2. Your insight is correct: When AlphaFold2 is used for computing sc-TM, the variance of TM is reduced than that of using ESMFold.
> > >
> > > Thanks again for your thorough comments and constructive suggestions!
> > >
> > > Best,
> > >
> > > Authors.

---

> > > ### Author Response · Authors · 2023-08-28
> > > **ProteinMPNN dataset uploaded**
> > >
> > > Dear reviewer,
> > >
> > > We have uploaded all the datasets into the following downable link:
> > >
> > > https://www.idrive.com/idrive/sh/sh?k=p9b2y3l6i5
> > >
> > > It is notable that due to the large size of ProteinMPNN dataset, we separated it into another file named mpnn.tar.gz.
> > >
> > > Thanks again for your careful review and valuable suggestions.
> > >
> > > Best,
> > >
> > > Authors

---

### Official Review · Reviewer_HGkS · 2023-07-03
**ProteinBench: Benchmarking Protein Design on Diverse Tasks, Models, and Metrics**

**Rating:** 8
**Confidence:** 4
**Correctness:** The work is generally correct.
**Clarity:** The manuscript is well written and ea…

**Strengths:**

+ Three datasets for protein design (CATH 4.3, PDB, and CASP15 de novo proteins) are provided.
+ Multiple open-source protein design models are included into the benchmark framework.
+ New/complementary metrics to evaluate protein design are added.
+ scTM and robustness can be a useful metric to evaluate the quality of designed protein sequences.
+ Several protein design methods are included in this benchmark. They were retrained and evaluated.
+ The evaluation results in this work can serve as a baseline for the development of new methods.



**Additional Feedback:**

N/A

**Documentation:**

The GitHub website provides a good document of the software package.

**Ethics:**

There is no ethics concern.

**Limitations:**

-	The confidence evaluation metric uses the average of prediction probabilities of the residues of a designed protein sequence to evaluate its quality. However, this metric can be method-dependent and may not be always applicable for method comparison. Some additional processing on the metric may be needed.
-	Some cutting-edge protein design methods such as RF Diffusion are not discussed and included.


**Opportunities For Improvement:**

It would be useful to compare scTM-score with the standard approach of calculating the TM-score of the predicted structure of the designed proteins against the experimental structures used as input. How similar or different are they? Is it possible to compare the two metrics on some designed proteins whose structures are known?

**Relation To Prior Work:**

Some recent protein design methods, particularly diffusion models, are not discussed.

**Summary And Contributions:**

This manuscript reports the first comprehensive benchmark for training and benchmarking protein design methods. Three complimentary datasets covering single chain proteins, multi-chain proteins, and de novo proteins are provided. Multiple complimentary evaluation metrics are supported. Several open-source protein design methods are included, which were retrained and evaluated by the metrics. Overall, the benchmark is a useful resource for users to train and test protein design methods.

---

> ### Author Response · Authors · 2023-08-17
> **Author rebuttal**
>
> Dear reviewer,
>
> Thanks for your valuable suggestions! Your comments will be helpful to help us improve the manuscript.
>
> > **Q1** The confidence evaluation metric uses the average of prediction probabilities of the residues of a designed protein sequence to evaluate its quality. However, this metric can be method-dependent and may not be always applicable for method comparison. Some additional processing on the metric may be needed.
> >> **Reply to Q1** Thanks for your valuable suggestions! Given that all methods utilize linear layers for predicting a softmax probability, it is indeed reasonable to conduct such a comparison. Alternatively, we could use the ratio of the maximum probability to the second-highest probability. We are happy and sincere to hear your suggestions.
>
> > **Q2** Some cutting-edge protein design methods such as RF Diffusion are not discussed and included.
> >> **Reply to Q2** We would like to clarify that we focus on the problem of protein inverse folding. The protein structure is known in our methods, while RF Diffusion is a structural generative model.
>
> We appreciate all the efforts you have made to assist us in improving our manuscript.
>
> Best,
>
> Authors.

---

### Official Review · Reviewer_oPQ8 · 2023-07-08
**Valuable benchmarks for inverse protein folding**

**Rating:** 8
**Confidence:** 4
**Correctness:** The claims are supported by experiments.
**Clarity:** The paper is well written.

**Strengths:**

1. The work is well-motivated. Deep learning bears the promise of solving protein inverse folding. However, it lacks comprehensive benchmarks to evaluate the performance of current and future models.
2. The proposed metrics, especially sc-TM, provide an important evaluation of inverse folding other than recovery rate alone.
3. ProteinBench covers a wide range of deep learning based inverse folding models as baselines.
4. The work extends the current database (i.e., CATH4.2) to include more diverse and challenging tasks including CATH4.3, PDB, and CASP15.
5. The authors provide a well-organized codebase to reproduce ProteinBench.

**Additional Feedback:**

N/A

**Documentation:**

Sufficient details on data collection and organization, availability and maintenance are included.

**Ethics:**

From my perspective, there aren't any ethical concerns that need further review.

**Limitations:**

The authors have discussed potential limitations adequately.

**Opportunities For Improvement:**

1. Though ProteinBench includes most deep learning based inverse folding models. It would be better to also benchmark physics-based methods (e.g., Rosetta) to better evaluate the performance of learning-based these models.
2. The confidence score (Conf) measures the predictive probability of AA sequences. In Tables 1&2, the authors annotate Conf as the higher the better, which may not be the case. As the model can be confident in predicting the wrong AA sequences. The authors may further clarify the metric.
3. The authors implement EMSFold to calculate sc-TM. It would be even better to include more protein folding models, like AlphaFold and OpenFold, to predict and compare structures from both predicted and ground-truth sequences. Since there are cases where one folding model fails.
4. Current ProteinBench extracts data from PDB/CASP15 to create multi-chain/de novo design tasks. These databases are dynamic and new data are added. The authors are suggested to discuss plans to maintain and extend ProteinBench with new data.
5. It can also be interesting to investigate how the baseline models perform if scaling up the number of parameters. For example, PiFold achieves second-best performance in most benchmarks but has much fewer parameters than the best-performing KWDesign.
6. Minor issues:
    - In line 189, TS45 is missing.
    - In Tables 1&2, the length should be $100 \leq L < 300$.

**Relation To Prior Work:**

The authors clearly discussed how this work differs from previous contributions.

**Summary And Contributions:**

In this work, the authors introduce ProteinBench, which aims at benchmarking the protein inverse folding problem. To this end, the authors present three components: datasets, models, and metrics. ProteinBench includes single-chain design tasks from CATH4.3, multi-chain design tasks from PDB, and de novo design tasks from CASP15, going beyond the widely used CATH4.2 database. It also implements multiple deep learning based models, e.g., ProteinMPNN, PiFold, and KWDesign. Lastly, it proposes evaluation metrics including diversity, sc-TM, confidence, robustness, and efficiency, which more comprehensively measure the performance of inverse folding models.

---

> ### Author Response · Authors · 2023-08-17
> **Author rebuttal**
>
> Dear reviewer,
>
> Thanks for your valuable suggestions! Your comments will be helpful to help us improve the manuscript.
>
> > **Q1** Though ProteinBench includes most deep learning based inverse folding models. It would be better to also benchmark physics-based methods (e.g., Rosetta) to better evaluate the performance of learning-based these models.
> >> **Reply to Q1** We plan to benchmark physics-based methods in the future, as these methods typically require the help of biological experts.
>
> > **Q2** The confidence score (Conf) measures the predictive probability of AA sequences. In Tables 1&2, the authors annotate Conf as the higher the better, which may not be the case. As the model can be confident in predicting the wrong AA sequences. The authors may further clarify the metric.
> >> **Reply to Q2** Thanks for your insightful comments. To clarify, higher confidence generally correlates with higher recovery rates. Taking PiFold and KWDesign as examples, we analyze recovery statistics within specific confidence ranges. In the test set of CATH4.2, we record the confidence and recovery (0 or 1) for each residue in all proteins. Finally, we calculate the average recovery for residues falling within a particular confidence range, shown as follows:
>
> >>| conf        | >0.0   | >0.1   | >0.2   | >0.3   | >0.4   | >0.5   | >0.6   | >0.7   | >0.8   | >0.9   | >0.95  |
> >>|----------|-------|-------|-------|-------|-------|-------|-------|-------|-------|-------|-------|
> >>| PiFold   | 51.58 | 51.60 | 55.43 | 63.04 | 69.87 | 75.99 | 81.75 | 86.66 | 90.96 | 95.09 | 97.44 |
> >>| KWDesign | 60.25 | 60.26 | 62.08 | 67.16 | 72.71 | 78.00 | 83.08 | 87.63 | 91.76 | 95.49 | 97.48 |
>
> >> The table shows that higher confidence generally correlates with higher recovery rates.
>
> > **Q3** The authors implement EMSFold to calculate sc-TM. It would be even better to include more protein folding models, like AlphaFold and OpenFold, to predict and compare structures from both predicted and ground-truth sequences. Since there are cases where one folding model fails.
> >> **Reply to Q3** Thanks for your valuable suggestions. The process of MSA searching incurs significant computational and time costs when utilizing AlphaFold and OpenFold. Currently, we are in the process of employing AlphaFold2 to evaluate the scTM. Once this analysis is complete, we will update the paper with the results obtained from AlphaFold2.
>
> > **Q4** Current ProteinBench extracts data from PDB/CASP15 to create multi-chain/de novo design tasks. These databases are dynamic and new data are added. The authors are suggested to discuss plans to maintain and extend ProteinBench with new data.
> >> **Reply to Q4** Thanks for your valuable suggestions! We plan to update ProteinBench when the CATH dataset (every 12 months) or the CASP competition (every 24 months) is updated.
>
> > **Q5** It can also be interesting to investigate how the baseline models perform if scaling up the number of parameters. For example, PiFold achieves second-best performance in most benchmarks but has much fewer parameters than the best-performing KWDesign.
> >> **Reply to Q5** On CATH4.2, we enlarge the PiFold model as follows:
>
>
> |                  | PiFold_base | Model1 | Model2 | Model3 | Model4 | Model5 | Model6 | Model7 | Model8 | KWDesign |
> |:----------------:|:-----------:|:------:|:------:|:------:|:------:|:------:|:------:|:------:|:--------:|:--------:|
> |     GNN Layer    |      10     |   10   |   10   |   10   |   20    |   15   |   20   |   30   |   40   |        |
> |      GNN Dim     |     128     |   256  |   512  |  1024  |   1024  |   128  |   128  |   128  |   128  |       |
> | Trainable Params |    5.79MB         |   22.70MB     |   90.15MB     |  359.51      |   716.41    |    8.60    |   11.41MB     |   17.03MB     |   22.64MB     |  41.67MB |
> |     Recovery     |    51.66         |   51.95     |   50.78     |  Out of memory      |   Out of memory     |   51.87     |   51.61     |   52.16     |   51.87     |   60.77  |
> |    Perpllexity   |    4.55         |  4.51      |   4.57     |        |        |   4.50     |   4.51     |   4.51     |   4.48     |   3.46   |
> >> We conclude that it may be difficult to further improve performance by simply enlarging PiFold.
>
> > **Q6** Minor issues: In line 189, TS45 is missing. In Tables 1&2, the length should be $100 \leq L < 300$
> >> **Reply to Q6** Thanks for your reminder. We have revised the paper accordingly.
>
> We appreciate all the efforts you have made to assist us in improving our manuscript.
>
> Best,
>
> Authors.

---

> > ### Comment · Reviewer_oPQ8 · 2023-08-26
> > **Response to rebuttal**
> >
> > I appreciate the authors' efforts in answering my questions. The authors have addressed most of my concerns and discussed future plans to further extend and maintain the benchmark. Therefore, I have increased my score by 1.

---

> > > ### Author Response · Authors · 2023-08-28
> > > **Further clarification about Q3**
> > >
> > > Dear reviewer,
> > >
> > > Thank you for your thoughtful review, constructive suggestions, and the improved score!  We provide further clarifications in terms of Q3.
> > >
> > > > **Q3**  The authors implement EMSFold to calculate sc-TM. It would be even better to include more protein folding models, like AlphaFold and OpenFold, to predict and compare structures from both predicted and ground-truth sequences. Since there are cases where one folding model fails.
> > > >> **Additional reply to Q3** As promised, we have updated Figure 3 with the results obtained from AlphaFold2. As hypothesized by reviewer Gd38, the utilization of AlphaFold2 yields more robust results, as it reduces the variance of the TMScore compared to using ESMFold.
> > >
> > > Thanks for your review service again!
> > >
> > > Best,
> > >
> > > Authors.

---

### Official Review · Reviewer_xYTD · 2023-07-20
**Solid evaluation framework for structure-based protein design**

**Rating:** 7
**Confidence:** 3
**Correctness:** Yes, the paper seems very correct.

**Strengths:**

The paper tests many structure-based models that are all cutting edge research tools. The evaluation criteria is clear and thorough.

**Additional Feedback:**

Good start to a unified protein design framework!

**Clarity:**

The paper is well written for someone in the field who is familiar with the work. However, there are some acronyms (i.e. 'sc-TM') that are never defined though the concepts behind them are illustrated in the body of the paper.

**Documentation:**

This paper pulls datasets from CATH4.3, PDB (same set as the proteinMPNN paper), and CASP15.

**Ethics:**

No, this all seems ethical

**Limitations:**

There are no obvious limitations of this work.

**Opportunities For Improvement:**

I wonder if there are much simpler models that could be used as a baseline with which to compare the newer models.

**Relation To Prior Work:**

Yes, figure one does a great job of at illustrating what is common / novel.

**Summary And Contributions:**

This submission creates an evaluation framework for protein structure designs. They train and test many different structure-based models and use metrics such as recovery, confidence, diversity, sc-TM, efficiency, and robustness to evaluate model performance. This is an important contribution to the field since there are not a standard set of evaluation criteria for these kinds of models.

---

> ### Author Response · Authors · 2023-08-17
> **Author rebuttal**
>
> Dear reviewer,
>
> Thanks for your valuable suggestions! Your comments will be helpful to help us improve the manuscript.
>
> > **Q1** The paper is well written for someone in the field who is familiar with the work. However, there are some acronyms (i.e. 'sc-TM') that are never defined though the concepts behind them are illustrated in the body of the paper.
> >> **Reply to Q1** Thanks for your comments! We would like to clarify that Eq.8 should be the definition of 'sc-TM'.
>
> > **Q2** I wonder if there are much simpler models that could be used as a baseline with which to compare the newer models.
> >> **Reply to Q2** We believe that PiFold is a much simpler baseline with competitive performance and state-of-the-art efficiency.
>
> We appreciate all the efforts you have made to assist us in improving our manuscript.
>
> Best,
>
> Authors.

---

### Official Review · Reviewer_wq41 · 2023-07-26
**nicely designed workflow and thorough benchmarks**

**Rating:** 7
**Confidence:** 3
**Correctness:** as far as this reviewer can tell.
**Clarity:** Yes

**Strengths:**

Including (to my knowledge) all or most of the SOTA protein design tools is a robust benchmarking result for the broader community. Further, extending the benchmarks from single-chain to multi-chain prediction tasks also steps the work up a notch, as multi-chain applications are a challenging yet impactful area of development. The inclusion of benchmarks on multiple evaluation metrics is one of the paper's strongest contributions, as is the use of a broader number of datasets compared to previous work.

**Additional Feedback:**

NA

**Documentation:**

appears to be.

**Limitations:**

see above.

**Opportunities For Improvement:**

It is possible that some of these different protein design tools were designed for specific applications, which would make the benchmarks on some of these tasks an unfair comparison. The authors should discuss this possibility. For many of the most critical metrics, the authors' own models perform the best (i.e. PiFold and KWDesign). For transparency, the authors could be sure to emphasize this in the main text and perhaps discuss why different tools perform better/worse in the different metrics.

**Relation To Prior Work:**

While some prior work is referenced, a more complete section on prior work would be helpful.

**Summary And Contributions:**

This paper presents a new workflow for benchmarking protein design tools. Protein design is currently a critically-important field of research, with impacts ranging from health/therapeutics to global sustainability. This paper points out that current methods of comparing the performance of different protein design tools are inadequate. To try and help solve this issue, they built ProteinBench, which sets a series of benchmarks across most of the latest open source protein design tools.

---

> ### Author Response · Authors · 2023-08-17
> **Author rebuttal**
>
> Dear reviewer,
>
> Thanks for your valuable suggestions! Your comments will be helpful to help us improve the manuscript.
>
> > **Q1** It is possible that some of these different protein design tools were designed for specific applications, which would make the benchmarks on some of these tasks an unfair comparison. The authors should discuss this possibility. For many of the most critical metrics, the authors' own models perform the best (i.e. PiFold and KWDesign). For transparency, the authors could be sure to emphasize this in the main text and perhaps discuss why different tools perform better/worse in the different metrics.
> >> **Reply to Q1** Thanks for your comments! In the field of AI-guided protein design, these approaches are fairly compared in the literature [1-7] on the CATH4.2 dataset. Here is an analysis of the relationships between different methods:
> >> - StructGNN and GraphTrans [1] use the C-alpha for geometric feature construction and achieves 35% recovery.
> >> - GCA [2] incorporates global attention into StructGNN, achieving 37% recovery using only C-alpha coordinates.
> >> - GVP [3] introduces a new type of GNN layer that learns invariant and equivariant features, achieving 39% recovery using only C-alpha coordinates.
>
> >> Up to now, previous methods suffer from poor inference efficiency due to autoregressive decoders.
>
> >> - AlphaDesign [4] introduces angle features, improves the GNN layer, and achieves 42% recovery. It replaces the autoregressive decoder with an iterative 1D CNN, significantly improving efficiency. AlphaDesign only uses C-alpha coordinates.
>
> >> Recent works significantly improve the recovery by using more knowledge, such as new types of atom coordinates or pretrained models.
>
> >> - ProteinMPNN [5] incorporates additional structural information, including C, N, O, and C-alpha coordinates. It modifies the GNN by adding an edge updating layer, resulting in 45% recovery.
> >> - PiFold [6] combines the strengths of AlphaDesign and ProteinMPNN, achieving 52% recovery. It simplifies the autoregressive decoder as a one-shot predictor, utilizes multi-scale residue interaction GNN layers, and uses C, N, O, and C-alpha coordinates.
> >> - KWDesign [7] is an ensemble model that combines PiFold with pretrained ESM models and acheives 61% recovery. It utilizes PiFold to create a prompt template for designed protein sequences. While the template may not be perfect, it can be refined by incorporating pre-trained knowledge. Both sequence pretraining (ESM-650M [8]) and structure pretraining models (ESMIF's encoder [9]) are used for extracting external knowledge.
>
> >> We have revised the paper and briefly discussed the relevance of these methods in the main text (line 118-126), but due to limited space, we cannot expand on the details. We recommend that the reader read the original papers.
>
>
> [1] Ingraham, John, et al. "Generative models for graph-based protein design." Advances in neural information processing systems 32 (2019).
>
> [2] Tan, Cheng, et al. "Generative de novo protein design with global context." arXiv preprint arXiv:2204.10673 (2022).
>
> [3] Jing, Bowen, et al. "Learning from protein structure with geometric vector perceptrons." arXiv preprint arXiv:2009.01411 (2020).
>
> [4] Gao, Zhangyang, Cheng Tan, and Stan Z. Li. "Alphadesign: A graph protein design method and benchmark on alphafolddb." arXiv preprint arXiv:2202.01079 (2022).
>
> [5] Dauparas, Justas, et al. "Robust deep learning–based protein sequence design using ProteinMPNN." Science 378.6615 (2022): 49-56.
>
> [6] Gao, Zhangyang, Cheng Tan, and Stan Z. Li. "PiFold: Toward effective and efficient protein inverse folding." arXiv preprint arXiv:2209.12643 (2022).
>
> [7] Gao, Zhangyang, Cheng Tan, and Stan Z. Li. "Knowledge-Design: Pushing the Limit of Protein Deign via Knowledge Refinement." arXiv preprint arXiv:2305.15151 (2023).
>
> [8] Lin, Zeming, et al. "Language models of protein sequences at the scale of evolution enable accurate structure prediction." BioRxiv 2022 (2022): 500902.
>
> [9] Hsu, Chloe, et al. "Learning inverse folding from millions of predicted structures." International Conference on Machine Learning. PMLR, 2022.
>
> We appreciate all the efforts you have made to assist us in improving our manuscript.
>
> Best,
>
> Authors.

---

> > ### Comment · Reviewer_wq41 · 2023-08-29
> > **Response to rebuttal**
> >
> > I appreciate the authors thorough explanations here, and additional edits to the paper. I maintain my score of good and recommendation for acceptance.

---

> > > ### Author Response · Authors · 2023-08-29
> > > **Thank you!**
> > >
> > > Thanks for your appreciation and review service! Your valuable suggestions have helped us improve this work!

---

### Decision · Program_Chairs · 2023-09-22

**Decision:**

Accept (Poster)

**Comment:**

The reviewers are enthusiastic about this work and consider it is a useful resource for users to train and test protein design methods, a solid evaluation framework for structure-based protein design, and a valuable benchmark for inverse protein folding. The review comments are also well addressed.